# Large language model detects previously undiagnosed heart failure with preserved ejection fraction in patients with metabolic-associated fatty liver disease: A multicenter cohort study

Xiaodan Lu[1☯], Xuliang Wang[2☯], Da Liu[3☯], Zhiyuan Zhang[4], Yuqiang Zhang[5], Tiancheng Zhang[5], Di Lu[1], Shengli Kuang[1], Chunjin Lai[6], Fushi Piao[7], Min Lin [8], Shengsong Zhu[9], Zhuchang Tian[10], Wei Fu[10], Yongbin Dai[3], Chengli Yao[6], Gang Li[11], Zhenzhong Zheng[3]*, Xiangbin Meng[5,6]*, Hongxing Luo [12,13]*

**1** Department of Gastroenterology, Henan Provincial People's Hospital, Zhengzhou University People's Hospital, Zhengzhou, China, **2** Department of Cardiology and Institute of Vascular Medicine, Peking University Third Hospital, Beijing, China, **3** Department of Cardiology and Institute of Vascular Medicine, Shenzhen Third People's Hospital, Shenzhen, China, **4** Department of Plastic and Burn Surgery, National Key Clinical Construction Specialty, The Affiliated Hospital of Southwest Medical University, Luzhou, Sichuan, China, **5** Peng Cheng Laboratory, Shenzhen, China, **6** Department of Cardiology, Dongguan Taixin Hospital, Dongguan, China, **7** Department of Cardiology, Peking University Shenzhen Hospital, Shenzhen, Guangdong, China, **8** Division of Cardiac Arrhythmia, Cardiac and Vascular Center, The University of Hong Kong-Shenzhen Hospital, Shenzhen, China, **9** Department of Cardiology, State Key Laboratory of Cardiovascular Disease, Fuwai Hospital, National Center for Cardiovascular Diseases, Chinese Academy of Medical Sciences and Peking Union Medical College, Beijing, China, **10** Department of Cardiology, The First Hospital of Hebei Medical University, Shijiazhuang, China, **11** Outpatient Department Office, Tongji Hospital Affiliated to Tongji Medical College of Huazhong University of Science and Technology, Wuhan, China, **12** Institute for Surgical Research, Oslo University Hospital, Rikshospitalet, Oslo, Norway, **13** Department of Physiology, Cardiovascular Research Institute Maastricht (CARIM), Maastricht University, Maastricht, the Netherlands

☯ These authors contributed equally to this study.
* h.luo@maastrichtuniversity.nl (HL); chinamengxiang@gmail.com (XM); greateful@163.com (ZZ)

## Abstract

Metabolic-associated fatty liver disease (MAFLD) and heart failure with preserved ejection fraction (HFpEF) share overlapping metabolic and inflammatory pathways, yet HFpEF is frequently underrecognized due to atypical symptoms and complex etiology. We aimed to evaluate a domain-tuned large language model (MedGuide-14B) for HFpEF detection from electronic health records (EHRs) among patients with MAFLD, and to compare outcomes between model-identified and clinically recognized cases. In this multicenter retrospective cohort, MedGuide-14B was fine-tuned on large-scale clinical encounters and utilized to analyze structured EHR data including demographics, comorbidities, and laboratory tests, together with free-text clinical notes. Patients were classified as clinically diagnosed HFpEF, MedGuide-identified HFpEF (defined as probability ≥0.70 based on ESC criteria), or non-HFpEF. Model performance was benchmarked against clinical diagnoses, and blinded validation was conducted for a prospectively sampled subset of MedGuide-identified cases. Outcomes included rehospitalization and mortality during follow-up. Among

**Data availability statement:** De-identified numerical data underlying the figures and tables in this study are publicly available at https://doi.org/10.5281/zenodo.17970844. These data include group-level baseline characteristics, model performance metrics, blinded clinician validation results, Kaplan–Meier survival curve values for all-cause and cardiovascular mortality, rehospitalization outcomes with unadjusted relative risks and odds ratios, calibration analysis data, and standardized inference prompts. Access to full patient-level data requires approval from the Institutional Review Board of Ethics Committees (contact: 15896850171@163.com).

**Funding:** This work was supported by the China Hebei International Joint Research Center for Structural Heart Disease, the Science and Technology Program of Hebei Province (Grant No. 22377719D to X.M.), the Hebei Province Finance Department Project (Grant No. LS202101 to X.M.), the Hebei Province Higher Education Science and Technology Research Project (Grant No. CXZX2025030 to X.M.), the Hebei Provincial Government–Funded Clinical Talent Project (Grant No. ZF2025062 to X.M.), the Research on Clinical Application of Medical Artificial Intelligence Program of the Hospital Management Institute of the National Health Commission (Grant Nos. YLXX24AIA008 and YLXX24AIA026 to X.M.), and the Shenzhen Third People's Hospital Research Fund (Grant No. 24252G1001 to Z.Z.). The funders had no role in study design, data collection and analysis, decision to publish, or preparation of the manuscript.

**Competing interests:** The authors have declared that no competing interests exist.

24,011 patients with MAFLD, 3,049 (12.7%) had clinically diagnosed HFpEF, while MedGuide-14B additionally identified 4,226 (17.6%) previously undiagnosed cases, of which 90.4% were confirmed on blinded validation (κ = 0.85). For clinically diagnosed HFpEF, model performance achieved an AUC of 0.94, with a sensitivity of 95.0% and a specificity of 92.3%. Rehospitalization occurred in 67.2% of clinically diagnosed HFpEF, 55.6% of MedGuide-identified HFpEF, and 38.4% of non-HFpEF patients (P < 0.001). At 48 months, cumulative all-cause mortality was 18.9%, 12.3%, and 4.6%, respectively, and cardiovascular mortality was 10.8%, 5.9%, and 1.5% (log-rank P < 0.05). Applied to routine EHR data, a domain-tuned large language model substantially increased the detection of HFpEF among patients with MAFLD, identifying a sizeable and previously unrecognized subgroup at intermediate yet clinically meaningful risk. Embedding such a model into EHR workflows may enable earlier evaluation and targeted testing, although prospective validation across diverse settings is warranted.

## Author summary

Heart failure with preserved ejection fraction (HFpEF) is a common and serious condition, but it is often difficult to recognize in routine clinical practice. Many patients have vague symptoms, and the information needed for diagnosis is scattered across electronic health records (EHRs). This problem is particularly pronounced in patients with metabolic-associated fatty liver disease (MAFLD) who are at increased cardiovascular risk but are not routinely evaluated for HFpEF.

In this multicenter study, we tested whether a medically aligned large language model, MedGuide-14B, could help identify previously unrecognized HFpEF by analyzing routine EHR data. The model reviewed both structured information (such as laboratory results and echocardiographic parameters) and free-text clinical notes without using raw imaging data.

Among more than 24,000 patients with MAFLD, MedGuide-14B identified over 4,200 additional patients who met guideline-based criteria for HFpEF but had not been clinically diagnosed. Most of these cases were confirmed by independent cardiologists, and their risks of rehospitalization and death were higher than those without HFpEF, though lower than patients already clinically diagnosed.

These findings suggest that large language models may support the earlier detection of HFpEF using existing clinical data. Such tools could assist clinicians by flagging high-risk patients for further evaluation, rather than replacing clinical judgment.

## Introduction

The global prevalence of metabolic-associated fatty liver disease (MAFLD) continues to rise, exerting a substantial impact on cardiovascular health through its strong association with multiple cardiometabolic disorders [1,2]. Heart failure with preserved ejection fraction (HFpEF) is a clinical syndrome characterized by typical heart failure symptoms and signs, preserved left ventricular ejection fraction (LVEF ≥ 50%), and objective evidence of structural or functional cardiac abnormalities accompanied by elevated natriuretic peptides or clinical congestion [3,4]. However, the diagnosis of HFpEF remains challenging owing to its atypical clinical presentation, multifactorial pathophysiology, and shared metabolic mechanisms with MAFLD, frequently leading to misdiagnosis or delayed recognition and, consequently, increased morbidity and mortality [5,6].

MAFLD and HFpEF share multiple pathogenic mechanisms, including insulin resistance, chronic systemic inflammation, and metabolic dysregulation, which collectively contribute to myocardial remodeling, microvascular dysfunction, and an increased risk of HFpEF [7,8]. In parallel, recent advances in artificial intelligence, particularly large language models (LLMs) such as ChatGPT, have demonstrated substantial promise in analyzing complex electronic health record (EHR) data to support timely and accurate clinical decision-making [9–11]. Through sophisticated natural language processing, LLMs can integrate heterogeneous structured variables and free-text clinical narratives, identify subtle or atypical disease patterns, and assist in early disease recognition and differential diagnosis [12–14]. MedGuide-14B is an open-source, medically aligned large language model fine-tuned on more than 260,000 de-identified real-world clinical encounters spanning multiple medical specialties. Rather than proposing a novel foundation model architecture, MedGuide-14B is designed to enhance clinical reasoning through domain-specific alignment and contextual understanding of medical language. We hypothesized that such a domain-tuned model could enable accurate screening for HFpEF among patients with MAFLD, a clinical context in which subtle symptoms and fragmented EHR signals are frequently overlooked during routine care.

Accordingly, this study aims to apply the MedGuide-14B to enhance the screening and detection of HFpEF among patients with MAFLD using routinely collected EHR data. By systematically comparing model-identified and clinically recognized HFpEF cases, we sought to quantify the extent of underdiagnosis and to assess the prognostic relevance of model-identified cases in terms of rehospitalization and mortality. More broadly, this work provides practical insights into how domain-tuned LLMs may be integrated into real-world clinical workflows to support earlier cardiovascular risk identification.

## Methods

### Study design and data sources

This retrospective multicenter cohort study included patients treated in outpatient and inpatient settings at Henan Provincial People's Hospital, Peking University Third Hospital, and the First Hospital of Hebei Medical University between February 2021 and January 2023. Data were extracted from EHRs, including demographic characteristics, comorbidities, laboratory tests, echocardiographic parameters, and longitudinal outcome information.

### Ethical considerations

The Ethics Committees of Henan Provincial People's Hospital, Peking University Third Hospital, and the First Hospital of Hebei Medical University approved this study in accordance with the Declaration of Helsinki (Ethics Approval Number: M2024271). Owing to the retrospective nature of the study and the use of de-identified data, the requirement for informed consent was waived by the aforementioned Ethics Committees.

### Inclusion and exclusion criteria

Inclusion criteria were defined a priori. Eligible patients were required to be ≥ 18 years of age and to have a diagnosis of MAFLD according to the 2020 International Expert Consensus Statement, defined by imaging evidence of hepatic

steatosis (ultrasound, computed tomography, or magnetic resonance imaging) in combination with one of the following: overweight or obesity (body mass index ≥25 kg/m$^2$ for Asians or ≥30 kg/m$^2$ for non-Asians) [15], type 2 diabetes mellitus, or at least two metabolic risk abnormalities. In addition, patients were required to have a documented left ventricular ejection fraction (LVEF) ≥50% on echocardiography within three months of MAFLD diagnosis and at least two liver function assessments (ALT, AST, GGT, or ALP) performed within the preceding year to ensure reliable phenotyping [16,17].

Exclusion criteria included incomplete key EHR variables (e.g., missing diagnostic documentation, imaging reports, liver function tests, or echocardiographic data). Patients with severe chronic conditions such as advanced malignancy, estimated glomerular filtration rate <30 mL/min/1.73 m$^2$, decompensated cirrhosis, or advanced cardiovascular disease were excluded. Additional exclusions applied to patients with acute conditions (e.g., acute coronary syndrome or acute infection) at baseline, as well as those who were unable to complete follow-up.

## HFpEF diagnosis criteria

The diagnosis of HFpEF was defined in accordance with European Society of Cardiology (ESC) guidelines [18]. Diagnostic criteria required the presence of heart failure-related symptoms and/or signs (including exertional dyspnea, fatigue, peripheral edema, pulmonary rales, jugular venous distension, hepatojugular reflux, or a third heart sound), a preserved left ventricular ejection fraction (LVEF ≥50%) on echocardiography with no prior documentation of reduced LVEF, and objective evidence of diastolic dysfunction [19]. Objective evidence was operationalized using established scoring systems, defined as either an H2FPEF score ≥6 or an HFA-PEFF score ≥5 [20,21].

## MedGuide-14B development and evaluation

MedGuide-14B is a domain-specific LLM developed through a customized medical alignment and fine-tuning program and is publicly available on Hugging Face (https://huggingface.co/clinic-research/MedGuide-14B). The model is based on a transformer architecture and was trained using the DeepSpeed framework on high-performance computing clusters. Supervised fine-tuning with cross-entropy loss, followed by reinforcement learning with human feedback (RLHF), was applied to align the model with clinical chain-of-thought reasoning and medical decision-making standards.

The model was further refined for medical vocabulary and clinical reasoning using more than 260,000 de-identified real-world clinical encounters, including structured EHR fields and free-text clinical notes across multiple medical specialties. Training data were obtained from participating hospitals under appropriate ethics approval. Supervised fine-tuning and expert-guided RLHF, involving clinician review and preference annotation, were integrated to improve consistency and clinical plausibility of model outputs. During inference, structured internal chain-of-thought reasoning strategies were employed to encourage coherent clinical decision-making, while only clinically relevant outputs and supporting evidence were exposed to users to mitigate the risk of spurious explanations. Additional details on the model invocation, standardized inference workflow, and probability thresholding strategy are provided in S1 File Method. An overview of the standardized inference workflow used for HFpEF identification, including data inputs, tokenization, transformer-based processing, and probability decoding, is illustrated in S1 Fig.

## Data processing

MedGuide-14B processed both structured and unstructured EHR data. Structured inputs included demographic variables, comorbidities, and laboratory measurements (e.g., ALT, AST, GGT, ALP, and natriuretic peptides), whereas unstructured inputs were derived from free-text clinical notes. Raw pixel-based imaging data from ultrasound, computed tomography, magnetic resonance imaging, or echocardiography were not analyzed directly. Instead, structured echocardiographic parameters (such as LVEF and E/e′) and textual interpretations from imaging reports were extracted and incorporated. For patients with MAFLD, hepatic steatosis was confirmed through radiology reports. For patients with HFpEF,

echocardiographic parameters including LVEF ≥50% and E/e′, together with diagnostic scores such as H2FPEF ≥6 or HFA-PEFF ≥5, were obtained either from structured EHR fields or from clinical notes parsed using natural language processing.

### Missing data handling

Preprocessing steps included text normalization to standardize clinical terminology and correct misspellings, imputation of missing laboratory and symptom variables using median (for continuous variables) or mode (for categorical variables), exclusion of records with missing key diagnostic variables, and integration of all available data into a unified representation suitable for model analysis.

### Patient classification & threshold selection

MedGuide-14B analyzed EHR data to generate a continuous probability score for HFpEF based on ESC diagnostic criteria, incorporating characteristic symptoms, LVEF ≥50%, and either an H2FPEF score ≥6 or an HFA-PEFF score ≥5. A probability threshold of 0.70 was selected based on optimization in a held-out validation dataset, balancing sensitivity and specificity in accordance with ESC priorities for HFpEF screening. This threshold was selected to favor sensitivity in a screening context, consistent with guideline-recommended case-finding strategies. Patients with MAFLD were consequently divided into three groups: (1) clinically diagnosed HFpEF group, (2) MedGuide-identified previously-undiagnosed HFpEF group, and (3) non-HFpEF group. Standardized prompts were consistently applied across all patient EHRs: "Identify patients meeting ESC HFpEF criteria (symptoms, LVEF ≥50%, H2FPEF ≥6 or HFA-PEFF ≥5) from EHR data, outputting a list of qualifying patients with supporting evidence (symptoms, echocardiographic parameters, scores)."

### Model inference and clinical interpretation

To enhance transparency and reproducibility, representative anonymized patient-level examples illustrating the full EHR input, MedGuide-14B inference process, handling of missing diagnostic elements, and final adjudicated interpretation are provided in the S3 File Method.

### Calibration analysis

Calibration performance was evaluated to assess the agreement between predicted probabilities and observed outcomes. Model calibration was quantified using the Brier score and visually examined with calibration plots constructed by grouping predicted probabilities into deciles. These analyses were performed in the validation dataset to complement discrimination metrics.

### Outcome data

MedGuide-14B was used to analyze undiagnosed HFpEF cases among enrolled MAFLD patients. Through in-depth EHR analysis, the model identified patients meeting ESC standards for HFpEF but not formally diagnosed.

Outcome measures included all-cause mortality, cardiovascular mortality, and rehospitalization rate during follow-up. Data were obtained via telephone follow-ups with patients or families and extracted from the EHR system. Follow-up assessments were conducted at regular intervals using a standardized protocol, combining telephone interviews and structured EHR review to ascertain rehospitalization and mortality outcomes.

### Statistical analysis

Continuous variables are presented as means with standard deviations, and categorical variables as frequencies with percentages. Distributional assumptions were assessed with the Shapiro-Wilk test. Normally distributed variables with

homogeneous variance, such as age and BMI, were compared using independent-sample *t* tests, whereas skewed or heteroscedastic variables, including E/e′ and natriuretic peptides, were analyzed with Mann-Whitney U tests. Categorical variables such as comorbidities were compared with *chi*-square or Fisher's exact tests.

Survival curves for rehospitalization and mortality were estimated using the Kaplan-Meier method, with differences between groups assessed by the log-rank test. For incidence estimates, confidence intervals were derived using the Wilson score method. Relative risks were log-transformed, and odds ratios were calculated using Woolf's method. All analyses were two-sided, and statistical significance was defined as $P<0.05$.

## Results

### Patient selection

A total of 24,878 patients with MAFLD were initially screened according to predefined inclusion and exclusion criteria. After excluding individuals with incomplete key records, severe comorbidities, acute conditions at baseline, or loss to follow-up, 24,011 patients remained for the final analysis. The stepwise selection process, including reasons for exclusion and subgroup allocation, is illustrated in Fig 1.

### Baseline characteristics

Among the 24,011 participants included in the final analysis, 3,049 (12.7%) had clinically diagnosed HFpEF, 4,226 (17.6%) were newly identified by MedGuide-14B as HFpEF without a prior clinical diagnosis, and 16,736 (69.7%) were classified as non-HFpEF.

Compared with the MedGuide-identified HFpEF group, patients with clinically diagnosed HFpEF were older (60.5±8.2 *vs.* 57.3±7.9 years, $P<0.001$) and exhibited a greater burden of cardiometabolic comorbidities. Echocardiographic assessment demonstrated more pronounced diastolic dysfunction and structural remodeling in the clinically diagnosed HFpEF group, including higher E/e′ ratios (13.5±3.0 *vs.* 12.2±2.7, $P<0.001$), increased interventricular septal thickness

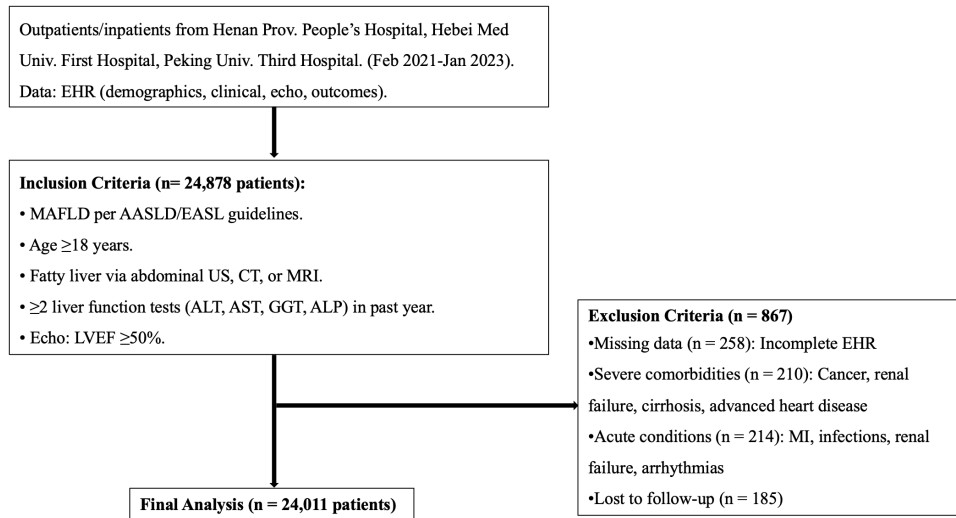

**Fig 1. Study flow diagram illustrating patient selection, exclusion criteria, and subgroup classification.** AASLD, American Association for the Study of Liver Diseases; ALP, alkaline phosphatase; ALT, alanine aminotransferase; AST, aspartate aminotransferase; CT, computed tomography; EASL, European Association for the Study of the Liver; EHR, electronic health record; Echo, echocardiography; GGT, gamma-glutamyl transferase; LVEF, left ventricular ejection fraction; MAFLD, metabolic dysfunction-associated fatty liver disease; MI, myocardial infarction; MRI, magnetic resonance imaging; US, ultrasound.

(12.2 ± 1.3 *vs.* 11.8 ± 1.3 mm, *P* < 0.001), larger left atrial dimensions (42.1 ± 4.0 *vs.* 40.3 ± 3.9 mm, *P* < 0.001), and slightly lower left ventricular ejection fraction (55.2 ± 4.8% *vs.* 56.5 ± 5.0%, *P* < 0.001).

In contrast, the non-HFpEF group represented the youngest and leanest population, characterized by the lowest estimated filling pressures (E/e′ = 10.5 ± 2.3, P < 0.001) and the highest left ventricular ejection fraction (58.5 ± 5.3%, P < 0.001). Detailed baseline characteristics are summarized in **Table 1**.

### Retrospective validation in clinically diagnosed HFpEF

When applied to patients with clinically established HFpEF, MedGuide-14B correctly identified 2,897 of 3,049 cases, achieving an area under the receiver operating characteristic curve (AUC) of 0.94 (95% CI: 0.92–0.96), with a sensitivity of 95.0% (95% CI: 94.2–95.8%) and a specificity of 92.3% (95% CI: 91.5–93.1%). The corresponding positive predictive value was 78.6% and the negative predictive value was 98.2% (**Table 2**).

**Table 1. Baseline demographic, clinical, and echocardiographic characteristics of the study population.**

| Variables | Total (n = 24,011) | Non-HFpEF (n = 16,736) | Clinical HFpEF (n = 3,049) | MedGuide HFpEF (n = 4,226) |
|---|---|---|---|---|
| Age, years | 54.3 ± 10.1 | 52.1 ± 9.0 | 60.5 ± 8.2[†] | 57.3 ± 7.9[†] |
| Male, n (%) | 15175 (63.2) | 10,871 (65.0) | 1,768 (58.0) [†] | 2,536 (60.0) [†] |
| BMI (kg/m²) | 28.7 ± 4.2 | 28.1 ± 4.0 | 30.2 ± 4.5[†] | 29.5 ± 4.3[†] |
| Systolic blood pressure (mmHg) | 132.8 ± 15.6 | 131.0 ± 15.5 | 136.0 ± 15.0[†] | 134.2 ± 15.4* |
| Diastolic blood pressure (mmHg) | 81.4 ± 10.4 | 80.5 ± 10.5 | 83.0 ± 9.8[†] | 82.1 ± 10.2* |
| Heart rate (bpm) | 72.5 ± 9.8 | 71.2 ± 9.9 | 74.3 ± 9.5[†] | 73.1 ± 9.7* |
| Diabetes mellitus, n (%) | 5878 (24.5) | 3,748 (22.4) | 918 (30.1) [†] | 1,212 (28.7) [†] |
| Hypertension, n (%) | 9762 (40.7) | 6,341 (37.9) | 1,473 (48.3) [†] | 1,948 (46.1) [†] |
| Hyperlipidemia, n (%) | 7651 (31.9) | 4,892 (29.2) | 1,204 (39.5) [†] | 1,555 (36.8) * |
| Coronary artery disease, n (%) | 5144 (21.4) | 3,313 (19.8) | 805 (26.4) [†] | 1,026 (24.3) [†] |
| Arrhythmia, n (%) | 3333 (13.9) | 2,127 (12.7) | 534 (17.5) [†] | 672 (15.9) [†] |
| Renal insufficiency, n (%) | 2787 (11.6) | 1,723 (10.3) | 476 (15.6) [†] | 588 (13.9) [†] |
| COPD, n (%) | 1500 (6.2) | 937 (5.6) | 250 (8.2) [†] | 313 (7.4) [†] |
| Carotid plaque/stenosis, n (%) | 2285(9.5) | 1,323 (7.9) | 451 (14.8)[†] | 511 (12.1) [†] |
| Obesity, n (%) | 5407(22.5) | 3,387 (20.2) | 905 (29.7) [†] | 1,115 (26.4) [†] |
| **Echocardiography** | | | | |
| AAD (mm) | 32.6 ± 3.8 | 32.3 ± 3.9 | 33.0 ± 3.7 | 32.9 ± 3.8 |
| LAAD (mm) | 39.4 ± 4.1 | 38.1 ± 3.8 | 42.1 ± 4.0[†] | 40.3 ± 3.9[†] |
| IVSd (mm) | 11.6 ± 1.4 | 11.3 ± 1.2 | 12.2 ± 1.3[†] | 11.8 ± 1.3[†] |
| LVEF (%) | 57.8 ± 5.2 | 58.5 ± 5.3 | 55.2 ± 4.8[†] | 56.5 ± 5.0[†] |
| LVPWT (mm) | 10.9 ± 1.3 | 10.6 ± 1.2 | 11.3 ± 1.2* | 11.1 ± 1.2 |
| LVEDD (mm) | 51.3 ± 5.0 | 51.0 ± 5.1 | 52.0 ± 4.8* | 51.6 ± 4.9 |
| LVESD (mm) | 34.7 ± 4.4 | 34.0 ± 4.4 | 36.0 ± 4.3[†] | 35.2 ± 4.3 |
| LVEDV (ml) | 129.3 ± 19.6 | 127.0 ± 19.9 | 134.0 ± 18.5[†] | 131.2 ± 18.8 |
| LVESV (ml) | 53.1 ± 10.5 | 50.7 ± 10.6 | 58.0 ± 9.9[†] | 54.8 ± 10.0* |
| E/e' ratio | 11.2 ± 2.5 | 10.5 ± 2.3 | 13.5 ± 3.0[†] | 12.2 ± 2.7* |

Data are presented as mean ± SD or n (%), as appropriate. Comparisons were performed using one-way analysis of variance or chi-square tests. Given the descriptive nature of baseline comparisons, no formal adjustment for multiple testing was applied. † P < 0.001; * P < 0.05.

AAD, Ascending aortic diameter; COPD, chronic obstructive pulmonary disease; IVSd, interventricular septum thickness; LAAD, left atrial anteroposterior diameter; LVEDD, left ventricular end-diastolic diameter; LVEDV, left ventricular end-diastolic volume; LVEF, left ventricular ejection fraction; LVESD, left ventricular end-systolic diameter; LVESV, left ventricular end-systolic volume; LVPWT, left ventricular posterior wall thickness.

**Table 2. Diagnostic performance of MedGuide-14B for clinically diagnosed HFpEF.**

|  | Value | 95% CI |
|---|---|---|
| Detected cases(n.) | 2,897 | -- |
| Sensitivity | 95.0% | 94.2–95.8% |
| Specificity | 92.3% | 91.5–93.1% |
| Positive Predictive Value (PPV) | 78.6% | 77.0–80.2% |
| Negative Predictive Value (NPV) | 98.2% | 97.8–98.6% |
| AUC | 0.94 | 0.92–0.96 |

### Blinded validation of newly identified HFpEF

In a blinded validation subset of 500 MedGuide-identified patients without a prior HFpEF diagnosis, 452 (90.4%) were independently confirmed as having HFpEF by two experienced cardiologists, with a strong inter-rater agreement (Cohen's κ = 0.85). The small proportion of non-confirmed cases was primarily attributable to insufficient symptom documentation, borderline diagnostic scores, or alternative echocardiographic interpretations (**Table 3**). The blinded adjudication workflow and consensus procedures are described in detail in S2 File Method.

### Survival and rehospitalization outcomes

Over a mean follow-up of approximately 46 months, survival and rehospitalization outcomes demonstrated a clear separation across the three study groups. Kaplan–Meier analysis showed early and sustained divergence in all-cause mortality, with the highest risk observed in the clinically diagnosed HFpEF group, an intermediate risk in the MedGuide-identified HFpEF group, and the lowest risk in the non-HFpEF group (**Fig 2A**). A similar pattern was observed for cardiovascular mortality (**Fig 2B**).

Rehospitalization followed the same stepwise trend. During follow-up, two thirds of clinically diagnosed HFpEF patients experienced at least one rehospitalization, compared with just over 50% in MedGuide-identified HFpEF and fewer than 40% in non-HFpEF patients. Relative to the non-HFpEF reference, rehospitalization risk was elevated 1.75-fold in clinical HFpEF and 1.45-fold in MedGuide HFpEF, with corresponding odds ratios of 3.32 and 2.00 (**Fig 3**).

### Calibration results

Calibration was assessed using the Brier score and decile-based calibration plots in the validation dataset. The predicted probabilities showed a close agreement with observed HFpEF status across risk deciles. Detailed calibration procedures and bin-level assessments are described in S3 File Method.

**Table 3. Blinded validation results for MedGuide-identified HFpEF cases.**

|  | Value | 95% CI or % |
|---|---|---|
| Confirmed HFpEF cases(n.) | 452/500 |  |
| Sensitivity (%) | 90.4 | 87.5–93.3 |
| Inter-rater reliability | Cohen's kappa = 0.85 | -- |
| Non-confirmed cases(n.) | 48 |  |
| • Inadequate symptom documentation | 28 | 58.3 |
| • Subthreshold H2FPEF/ HFA-PEFF | 15 | 31.3 |
| • Discrepant echocardiography | 3 | 6.3 |
| • Competing diagnoses | 2 | 4.1 |

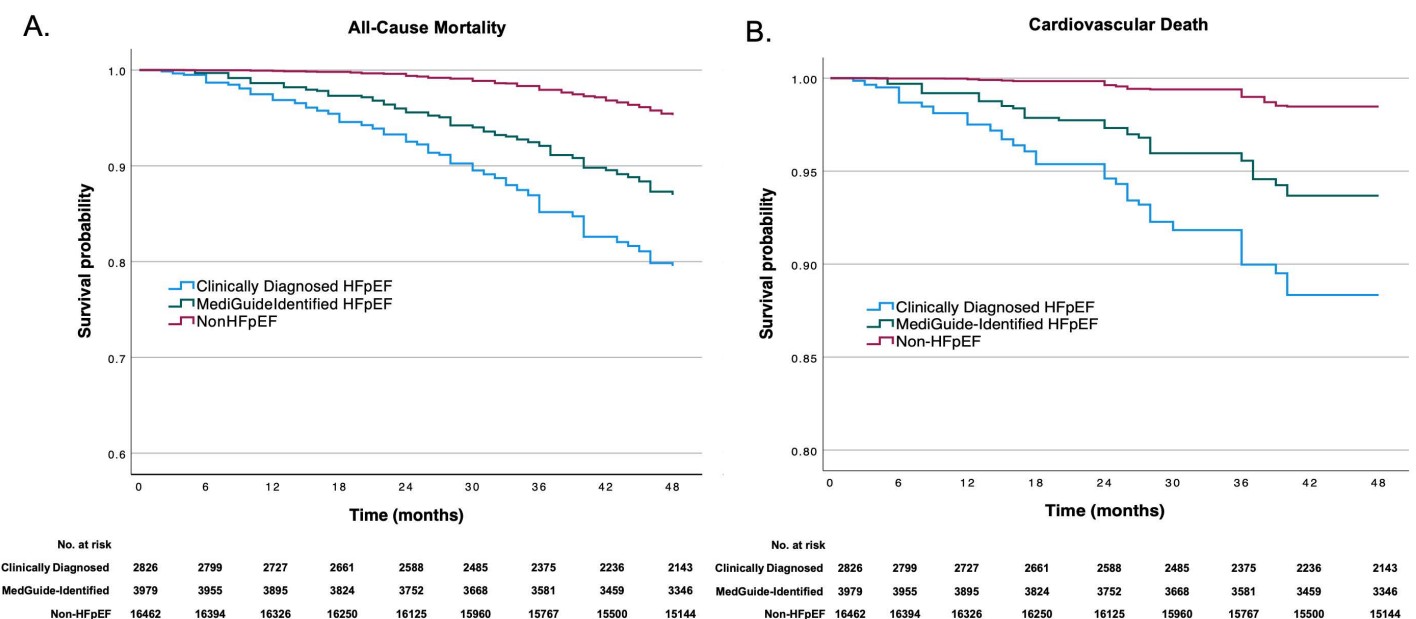

**Fig 2. Kaplan–Meier survival curves for mortality outcomes. (A)** All-cause mortality and **(B)** cardiovascular mortality among patients with non-HFpEF, clinically diagnosed HFpEF, and MedGuide-identified HFpEF. Survival probabilities were estimated using the Kaplan-Meier method, and differences between groups were assessed using the log-rank test. Analyses were unadjusted and based on group-level follow-up data.

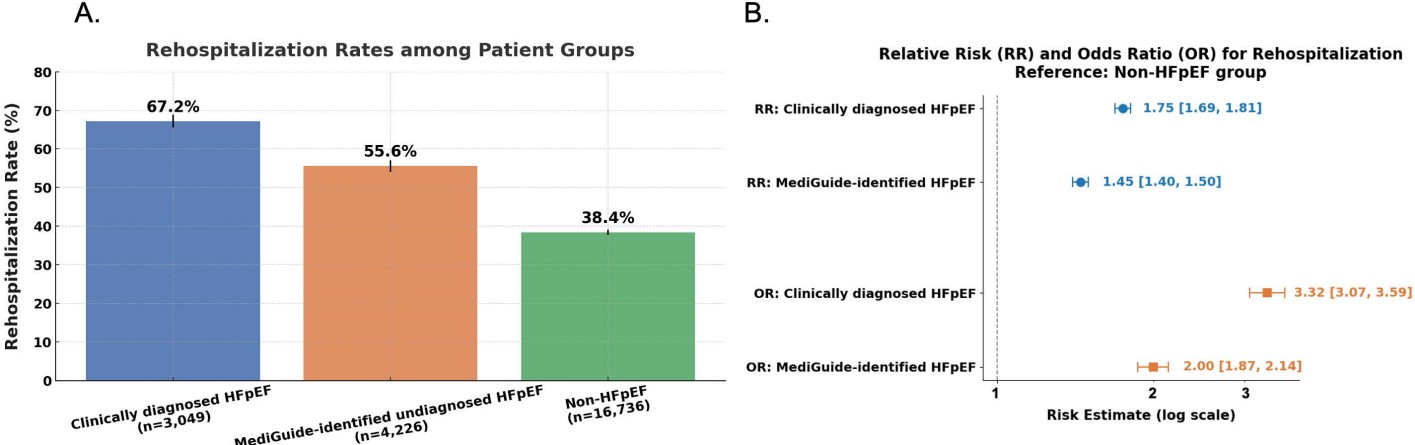

**Fig 3. Rehospitalization outcomes and unadjusted risk estimates across study groups. (A)** Crude rehospitalization rates in patients with clinically diagnosed HFpEF, MedGuide-identified HFpEF, and non-HFpEF. Bars represent the proportion of patients experiencing at least one rehospitalization during follow-up. **(B)** Unadjusted relative risks (RRs) and odds ratios (ORs) for rehospitalization, using the non-HFpEF group as the reference. Estimates were calculated based on group-level event counts without adjustment for covariates. RR and OR are shown to facilitate comparison with prior observational studies.

## Discussion

In this multicenter retrospective cohort study, we demonstrated that MedGuide-14B, a domain-tuned 14-billion-parameter LLM, can analyze routinely collected EHR data to identify HFpEF among patients with MAFLD with high diagnostic

accuracy. By additionally identifying more than 4,200 HFpEF cases beyond routine clinical recognition, the model reveals a sizeable and previously overlooked population at elevated risk of rehospitalization and mortality. Together, these findings suggest that medically aligned LLMs may provide a scalable approach to addressing diagnostic blind spots at the liver–heart interface when embedded into existing EHR workflows.

## HFpEF under-recognition in MAFLD and the role of EHR-based AI screening

HFpEF remains under-recognized in patients with MAFLD because its clinical presentation is often nonspecific, its etiology is multifactorial, and multiple metabolic drivers are shared between hepatic and cardiac dysfunctions [3,22]. Recent studies have consistently demonstrated a strong association between metabolic liver disease and HFpEF, while highlighting persistent challenges in timely recognition of HFpEF in these high-risk populations [23–28]. Advanced fibrosis in NAFLD/MAFLD has been associated with approximately 1.5–2-fold higher odds of HFpEF after adjustment for obesity and other cardiometabolic risk factors. Large real-world analyses further indicate that a substantial proportion of HFpEF cases remain clinically unrecognized. For example, Wu *et al*. reported that more than 75% of patients with a preserved ejection fraction (LVEF ≥50%) met ESC HFpEF diagnostic criteria despite lacking a formal diagnosis, and these patients experienced substantial long-term mortality.

Parallel efforts leveraging EHRs, imaging, and machine learning have begun to improve HFpEF identification. In obese or cardiometabolic cohorts, machine learning-based models integrating clinical and echocardiographic features have achieved discrimination performance exceeding traditional rule-based scores, underscoring the potential value of data-driven screening approaches in populations where HFpEF is frequently overlooked.

To place the current findings in context, Table 4 summarizes representative studies from the past five years that have investigated HFpEF identification in MAFLD or related cardiometabolic cohorts, highlighting differences in study populations, data modalities, methodological approaches, and outcome definitions.

In our multicenter cohort, the domain-tuned LLM MedGuide-14B integrated structured variables and free-text EHR narratives from 24,011 patients with MAFLD and identified 17.6% additional HFpEF cases that were not recognized during routine clinical care, highlighting the magnitude of underdiagnosis in this high-risk population.

These findings are supported by recent physiological mechanistic studies between liver diseases and HFpEF. Hepatic lipid accumulation triggers low-grade systemic inflammation and raises circulating IL-6, TNF-α and soluble ST2. These mediators promote myocardial fibrosis and microvascular rarefaction, both serving as the early hallmarks of HFpEF [29,30]. Qualitative examination of model outputs suggests that MedGuide-14B frequently emphasizes references to these biomarkers together with exertional dyspnea, indicating that the model captures the inflammatory liver-heart axis and synthesizes disparate signals into an early HFpEF prediction [31,32]. By embedding diagnostic protocols within a purpose-built LLM, MedGuide-14B translates guideline concepts into point-of-care decision support, augmenting clinical judgement in complex metabolic syndromes [33,34]. The model therefore offers a practical path toward precision hepato-cardiology, enabling integrated hepatic and cardiac information to trigger timely, personalized intervention. Representative anonymized patient-level examples illustrating the model's reasoning under complete and partially missing diagnostic information are provided in S4 File Method.

## Comparison with other LLMs and conventional machine learning

Compared with general-purpose LLMs, domain-tuned models such as MedGuide-14B may offer greater consistency for HFpEF screening tasks that require the integration of heterogeneous EHR data. While prior studies have demonstrated the potential of LLMs in medical reasoning and clinical decision support, their performance in specific diagnostic contexts can vary widely depending on domain alignment and input structure [35,36]. Our findings suggest that targeted medical alignment may enhance the reliability of LLM-based screening in complex cardiometabolic conditions. However, direct head-to-head comparisons with other foundation models were beyond the scope of this study.

**Table 4. Comparison of recent studies on HFpEF identification in MAFLD or related cardiometabolic cohorts.**

| Author (Year) | Cohort/ Setting | Data Modality | Method/ Model | HFpEF Outcome Definition | Performance Metrics | Follow-up | Key Findings/ Notes |
|---|---|---|---|---|---|---|---|
| Our Study (2025) | MAFLD patients at risk; multicenter cohort (n = 24,011) | EHR (structured variables + free-text clinical notes); echocardiographic reports* | Domain-tuned LLM (MedGuide-14B) | Guideline-based HFpEF (symptoms + echocardiographic criteria) | AUC 0.94 (95% CI: 0.92–0.96); sensitivity 95.0%; specificity 92.3%; PPV 78.6%; NPV 98.2% | Cross-sectional | First study applying a medically aligned LLM to HFpEF screening in a MAFLD population; identified a substantial number of HFpEF cases not recognized during routine care using multimodal EHR information. |
| Bermea et al. (2024) | Obese HFpEF vs. controls; single-center (n = 471) | Clinical data + echocardiography (EHR) | Machine learning (gradient boosting → logistic model) | Clinical HFpEF diagnosis in obese cohort | AUC 0.88; Sens 83%; Spec 82% | Retrospective | Developed HFpEF-JH score tailored to obesity; outperformed H2FPEF score but limited to high-BMI populations. |
| Wu et al. (2024) | HF patients with LVEF ≥50%; multicenter EHR cohort (UK; n = 3,727) | EHR notes + echocardiographic data | Rule-based NLP phenotyping (CogStack/MedCAT) | ESC HFpEF diagnostic criteria (HFA-PEFF) | Not applicable (phenotyping study) | Up to 5 years | Demonstrated marked underdiagnosis: 75.4% met HFpEF criteria without formal diagnosis; high long-term mortality; highlighted NLP as a screening tool rather than a predictive model. |
| Jiang et al. (2023) | T2DM inpatients with vs. without NAFLD; single-center (China; n = 2,418) | Laboratory fibrosis markers + echocardiography | Statistical analysis (logistic regression) | HFpEF defined by EF ≥ 50% + diastolic dysfunction | OR for advanced fibrosis = 1.59 | Cross-sectional | Advanced liver fibrosis independently associated with HFpEF; simple steatosis alone not predictive; no discrimination metrics reported. |
| Lee et al. (2020) | T2DM patients ≥50 years with ultrasound-defined NAFLD; single-center (Korea; n = 606) | Liver ultrasound + echocardiography | Statistical analysis (logistic regression) | LV diastolic dysfunction (echo-based) | OR for fibrosis ≈ 1.58 | Cross-sectional | NAFLD associated with subclinical diastolic dysfunction; advanced fibrosis showed strongest association with HFpEF-related phenotypes. |
| Chang et al. (2025) | MASLD vs. non-MASLD; multicenter health system cohort (Taiwan; n = 26,676) | EHR diagnoses, laboratory data + echocardiography | Cohort study (competing-risk regression) | Incident HF and HFpEF by echocardiographic sub-classification | SHR for HFpEF = 1.91 (MASLD vs non-MASLD) | Median 6 years | MASLD nearly doubled risk of HFpEF; risk increased with metabolic burden; emphasized need for proactive HFpEF surveillance. |
| Wegermann et al. (2025) | Biopsy-proven MASLD; prospective registry (USA; n = 570) | Liver biopsy + EHR (manual symptom, BNP, echo review) | Longitudinal observational analysis | Incident HF (clinical events or HF signs + biomarkers) | 18% incident HF over 11 years | Median 11 years | High HF incidence and underdiagnosis in MASLD; nearly half showed HF signs without diagnosis; most cases consistent with HFpEF phenotype. |

AUC, area under the curve; BMI, body mass index; BNP, B-type natriuretic peptide; CI, confidence interval; EHR, electronic health record; ESC, European Society of Cardiology; EF, ejection fraction; HF, heart failure; HFpEF, heart failure with preserved ejection fraction; HFpEF-JH, heart failure with preserved ejection fraction–Jiangsu/Hospital score; LLM, large language model; LVEF, left ventricular ejection fraction; LV, left ventricle/left ventricular; MAFLD, metabolic dysfunction-associated fatty liver disease; MASLD, metabolic dysfunction-associated steatotic liver disease; NAFLD, nonalcoholic fatty liver disease; NLP, natural language processing; NPV, negative predictive value; OR, odds ratio; PPV, positive predictive value; Sens, sensitivity; SHR, subdistribution hazard ratio; Spec, specificity; T2DM, type 2 diabetes mellitus.

Conventional machine learning models lacked generalizability for HFpEF screening [37]. Wu et al.'s natural language processing-based approach using CogStack and MedCAT achieved a sensitivity of approximately 85% but struggled with complex medical terminology and non-standard EHR data [24]. In contrast, MedGuide-14B's sensitivity (95.0%) and accuracy (AUC: 0.94) demonstrate higher predictive accuracy, leveraging transformer architecture to process diverse EHR

data effectively. Deep learning models using electrocardiogram data reported high sensitivity but were limited by reliance on specific data types and complex preprocessing [38].

### LLM-identified HFpEF is associated with worse patients' outcomes

To assess the prognostic relevance of MedGuide-14B, rehospitalization and survival were examined in the three study cohorts. The results revealed a clear gradient of risk. Patients identified exclusively by MedGuide-14B exhibited more favorable outcomes than those diagnosed through routine clinical practice, likely reflecting an earlier or less advanced disease stage at the time of detection. This interpretation is supported by their comparatively lower baseline H2FPEF scores and less severe echocardiographic abnormalities [39]. Nevertheless, the risks of rehospitalization and mortality in the MedGuide-identified group remained substantially higher than those observed in the non-HFpEF cohort, indicating that the model is capturing clinically meaningful disease rather than false-positive signals.

Independent adjudication by senior cardiologists affirmed the validity of most algorithm-flagged cases, and the small number of discordant assessments was attributable to incomplete symptom documentation or borderline diagnostic scores. These findings demonstrate that routine integration of a domain-tuned LLM can expose a sizeable subset of incipient HFpEF that conventional clinical workflows frequently overlook, thereby opening a window for earlier, potentially outcome-modifying intervention [40,41].

Although MedGuide-14B does not provide explicit feature attribution in the manner of traditional machine learning models, qualitative examination of model outputs suggests that it consistently leverages clinically relevant cues documented in EHRs. These include exertional dyspnea described in free-text notes, preserved LVEF with elevated filling pressures, and high H2FPEF or HFA-PEFF scores embedded within structured data. In several MedGuide-identified cases, such cues were present but fragmented across clinical notes and diagnostic reports, potentially contributing to their omission during routine care. This pattern illustrates how a domain-tuned LLM may synthesize dispersed clinical signals into a coherent risk assessment.

### Clinical integration to support non-cardiologist decision-making

Embedding MedGuide-14B within EHR systems could support front-line clinicians including gastroenterologists, primary care physicians, and non-cardiology specialists by generating automated alerts when combinations of structured data and unstructured clinical notes suggest possible HFpEF. Such alerts may prompt targeted diagnostic evaluation, such as natriuretic peptide testing or focused echocardiography, and facilitate timely referral to cardiology services [42,43]. Importantly, MedGuide-14B is intended to function as a decision-support tool rather than a diagnostic replacement, with final clinical judgment remaining with treating physicians.

### Limitations

Several limitations should be acknowledged. First, the retrospective design relies on routine EHR documentation and missing or inconsistently recorded data may have influenced both model inputs and outcome ascertainment. Second, all data were derived from three centers in China, which may limit the model generalizability to other healthcare systems, populations, and EHR architectures. Third, although MedGuide-14B demonstrated favorable discrimination and calibration, external validation in independent cohorts remains necessary before broad clinical deployment. Finally, successful real-world implementation will depend on clinician trust, intuitive user interfaces, and seamless integration with existing information systems.

### Conclusions

In summary, MedGuide-14B enables the identification of previously unrecognized HFpEF among patients with MAFLD by integrating structured and unstructured EHR data. The strong concordance with clinician review supports its feasibility as

a screening and decision-support tool, while further prospective and external validation will be essential to establish its role in routine clinical practice.

## Supporting information

**S1 Fig. Inference workflow of MedGuide-14B for HFpEF identification.** The figure illustrates the inference pipeline used in this study. Structured and unstructured electronic health record (EHR) data were provided as input to MedGuide-14B, which processed the information through tokenization, transformer-based contextual representation, and probabilistic inference. The model output a probability score representing the likelihood of heart failure with preserved ejection fraction (HFpEF). This schematic reflects the model invocation and inference process only and does not depict model training or fine-tuning procedures.
(TIFF)

**S1 File. Method. Supplementary methodological details.** This document provides additional methodological information not fully described in the main Methods section, including details on data preprocessing and normalization, handling of missing data, MedGuide-14B model invocation and inference workflow, standardized prompt structure, probability threshold selection for HFpEF classification, and supplementary statistical analyses.
(DOCX)

**S2 File. Method. Blinded validation adjudication protocol.** This document describes the blinded adjudication workflow and consensus procedures used for independent review of MedGuide-identified HFpEF cases, including reviewer selection criteria, case assignment, and discordance resolution.
(DOCX)

**S3 File. Method. Calibration assessment and representative patient-level inference examples.** This document presents detailed calibration procedures and bin-level assessment of MedGuide-14B predicted probabilities, along with representative anonymized patient-level examples illustrating the full EHR input, model inference process, handling of missing diagnostic elements, and final adjudicated interpretation.
(DOCX)

**S4 File. Method. Model reasoning under incomplete diagnostic data.** This document provides representative anonymized patient-level examples illustrating the model's diagnostic reasoning under conditions of complete and partially missing clinical information, demonstrating how MedGuide-14B handles uncertainty in real-world EHR data.
(DOCX)

## Acknowledgments

We thank Sirui Han, Zijia Zhu, and Mengshen Wang for their valuable assistance during the manuscript revision process, including editorial support, language refinement, and help in addressing reviewer comments. Their contributions are gratefully acknowledged.

## Author contributions

**Conceptualization:** Xiaodan Lu, Xuliang Wang, Shengsong Zhu.

**Data curation:** Xiaodan Lu, Xuliang Wang, Zhiyuan Zhang, Yuqiang Zhang, Xiangbin Meng.

**Formal analysis:** Xiaodan Lu, Yuqiang Zhang, Yongbin Dai.

**Funding acquisition:** Zhiyuan Zhang.

**Investigation:** Xuliang Wang, Yuqiang Zhang, Di Lu, Fushi Piao, Shengsong Zhu, Wei Fu, Hongxing Luo.

**Methodology:** Da Liu, Zhiyuan Zhang, Yuqiang Zhang, Tiancheng Zhang, Di Lu, Min Lin, Zhenzhong Zheng, Xiangbin Meng, Hongxing Luo.

**Project administration:** Da Liu, Zhiyuan Zhang, Di Lu, Min Lin, Hongxing Luo.

**Resources:** Da Liu, Fushi Piao, Wei Fu, Chengli Yao, Chunjin Lai, Gang Li, Hongxing Luo.

**Software:** Xiaodan Lu, Tiancheng Zhang, Fushi Piao, Shengsong Zhu, Zhuchang Tian, Yongbin Dai, Chengli Yao, Chunjin Lai, Gang Li, Zhenzhong Zheng, Xiangbin Meng, Hongxing Luo.

**Supervision:** Xuliang Wang, Da Liu, Tiancheng Zhang, Shengli Kuang, Min Lin.

**Validation:** Xuliang Wang, Da Liu.

**Visualization:** Da Liu, Tiancheng Zhang, Zhuchang Tian.

**Writing – original draft:** Xiaodan Lu, Xuliang Wang, Shengli Kuang, Xiangbin Meng.

**Writing – review & editing:** Xiaodan Lu, Xiangbin Meng, Hongxing Luo.

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
