## [Decision Letter · Decision Letter 0]

24 Nov 2025

Response to Reviewers '. This file does not need to include responses to any formatting updates and technical items listed in the 'Journal Requirements' section below.* A marked-up copy of your manuscript that highlights changes made to the original version. You should upload this as a separate file labeled 'Revised Manuscript with Track Changes '.* An unmarked version of your revised paper without tracked changes. You should upload this as a separate file labeled 'Manuscript '. If you would like to make changes to your financial disclosure, competing interests statement, or data availability statement, please make these updates within the submission form at the time of resubmission. Guidelines for resubmitting your figure files are available below the reviewer comments at the end of this letter. We look forward to receiving your revised manuscript. Kind regards, Thomas Kyumwa Kisimbi, RN, MA, MPA, MBAAcademic EditorPLOS Digital Health Amara TariqSection EditorPLOS Digital Health Leo Anthony CeliEditor-in-ChiefPLOS Digital Healthorcid.org/0000-0001-6712-6626  **Journal Requirements:**

1. Please provide an Author Summary. This should appear in your manuscript between the Abstract (if applicable) and the Introduction, and should be 150–200 words long. The aim should be to make your findings accessible to a wide audience that includes both scientists and non-scientists. Sample summaries can be found on our website under Submission Guidelines: [LINK]

https://journals.plos.org/digitalhealth/s/submission-guidelines#loc-parts-of-a-submission

**Additional Editor Comments (if provided):** The study is innovative and aligns closely with the journal’s aims, but it requires enhanced methodological transparency, calibration analysis, data openness, and clearer articulation of clinical implications. The results are promising yet preliminary without external validation or model interpretability evaluation. Addressing these issues would elevate the paper to a publishable standard.

Methodology

Strengths

• Well-defined inclusion/exclusion criteria based on ESC HFpEF diagnostic criteria.

• Appropriate selection of performance metrics (AUC, PPV, NPV, sensitivity, specificity).

• Clinical adjudication by cardiologists enhances reliability.

Concerns / Suggestions

• Clarify model architecture and training: Was MedGuide-14B trained exclusively on Chinese EHR data, and was it fine-tuned using domain-specific corpora (e.g., cardiology notes)?

• Describe training–validation–test data splits and strategies to prevent data leakage.

• Provide calibration performance (e.g., Brier score, calibration plots) to complement discrimination metrics.

• Discuss model interpretability: How are feature attributions or token contributions verified against clinical reasoning?

• Expand methods for handling missing or inconsistent EHR data.

• Consider external validation or cross-site holdout testing to confirm generalizability.

Data

The data-availability statement limits access to “reasonable request,” which is non-compliant with PLOS open-data policy. Derived or anonymized data, synthetic EHRs, and reproducible model code should be made publicly accessible to support transparency. I recommend providing:

• De-identified or synthetic EHR datasets sufficient for replication.

• Model code, prompts, or weights on a public platform (e.g., GitHub, Zenodo).

**Reviewers' Comments:** Reviewer's Responses to Questions

**Comments to the Author**

1. Does this manuscript meet PLOS Digital Health’s publication criteria ? Is the manuscript technically sound, and do the data support the conclusions? The manuscript must describe methodologically and ethically rigorous research with conclusions that are appropriately drawn based on the data presented.

Reviewer #1: Yes

Reviewer #2: Yes

Reviewer #3: Partly

2. Has the statistical analysis been performed appropriately and rigorously?

Reviewer #1: Yes

Reviewer #2: Yes

Reviewer #3: Yes

3. Have the authors made all data underlying the findings in their manuscript fully available (please refer to the Data Availability Statement at the start of the manuscript PDF file)?

Reviewer #1: No

Reviewer #2: Yes

Reviewer #3: Yes

4. Is the manuscript presented in an intelligible fashion and written in standard English?

Reviewer #1: Yes

Reviewer #2: Yes

Reviewer #3: Yes

Reviewer #2: This is an interesting study to evaluate the large language model MedGuide-14B for HFpEF detection. There are several suggestions to improve the manuscript:

1.It cannot be seen from the title that the patient has MAFLD. The title needs to be revised.

2.The first paragraph of "Ethical considerations" is not about ethical issues; it is about study design or data collection. I suggest to add the “study design” section.

3.In the “MedGuide-14B development and evaluation” section, it is better to report the amount of training data used.

4.Why was the threshold of 0.7 selected to classify patients as HFpEF-positive? What is the basis for choosing to use 0.7?

5.Since this is a cohort study, the method needs to clearly specify how the follow-up is conducted, how often it is carried out, and what contents are being followed. It should not be simply written in a brief manner “Data were obtained via telephone follow-ups with patients or families and extracted from the electronic health record system (PIMS)”.

6.All figures need to be made clearer.

7.Table 1: Did the Chi-square test consider the issue of multiple comparisons and perform Bonferroni adjustmen?

8.It is not recommended to divide Table 2 into (a) and (b) tables; instead, it is suggested to present it as two separate tables.

Reviewer #3: This paper investigates the application of MedGuide-14B, a domain-specific large language model, for the detection of heart failure with preserved ejection fraction (HFpEF) from electronic health records (EHRs) among patients with metabolic-associated fatty liver disease (MAFLD). MedGuide-14B is a purpose-built, open-source LLM, fine-tuned (with RLHF) on genuine multi-specialty medical dialogues and EHRs, not simply off-the-shelf.

I believe that the paper needs to further improve the following content to enhance its scientific influence:

1. More clearly indicate whether training and open-sourcing MedGuide-14B is a contribution of this paper, or whether applying MedGuide-14 to the detection of HFpEF is a contribution.

2. If considering the training and open-sourcing of MedGuide-14B as the main contribution, please supplement more detailed training details.The description of MedGuide-14B remains quite high-level. Please provide clearer details on fine-tuning, data preprocessing (for both structured variables and free text), prompt design, and inference setup.

3. I would encourage the authors to include several qualitative case studies comparing MedGuide-14B’s outputs with the corresponding clinical documentation and final diagnoses. In particular, presenting a small number of representative patients among the 4,226 newly identified HFpEF cases could illustrate which cues the model appears to leverage and why these may have been overlooked in routine care. Such case-level examples would make the findings more tangible and help elucidate why MedGuide-14B is able to identify additional cases beyond standard clinical practice.

4. Include a separate Literature Review section to includea more systematic and up-to-date overview of related work on large language models and other foundation models in cardiology and, more broadly, in clinical medicine.

5. For reproducibility, the manuscript should provide a clearer description of how MedGuide-14B was invoked (For example, the form of input and output, how it has assisted in the judgment of HFpEF) and how the experiments could be replicated by other groups.

**Do you want your identity to be public for this peer review?** For information about this choice, including consent withdrawal, please see our Privacy Policy .

Reviewer #2: No

Reviewer #3: No

**Figure resubmission:**  While revising your submission, we strongly recommend that you use PLOS’s NAAS tool (https://ngplosjournals.pagemajik.ai/artanalysis) to test your figure files. NAAS can convert your figure files to the TIFF file type and meet basic requirements (such as print size, resolution), or provide you with a report on issues that do not meet our requirements and that NAAS cannot fix. 

**Reproducibility:** To enhance the reproducibility of your results, we recommend that authors of applicable studies deposit laboratory protocols in protocols.io, where a protocol can be assigned its own identifier (DOI) such that it can be cited independently in the future. Additionally, PLOS ONE offers an option to publish peer-reviewed clinical study protocols. Read more information on sharing protocols at https://plos.org/protocols?utm_medium=editorial-email&utm_source=authorletters&utm_campaign=protocols To enhance the reproducibility of your results, we recommend that authors of applicable studies deposit laboratory protocols in protocols.io, where a protocol can be assigned its own identifier (DOI) such that it can be cited independently in the future. Additionally, PLOS ONE offers an option to publish peer-reviewed clinical study protocols. Read more information on sharing protocols at https://plos.org/protocols?utm_medium=editorial-email&utm_source=authorletters&utm_campaign=protocols

---

## [Decision Letter · Decision Letter 1]

5 Mar 2026

Large language model detects previously undiagnosed heart failure with preserved ejection fraction in patients with metabolic-associated fatty liver disease: a multicenter cohort study

PDIG-D-25-00681R1

Dear Dr. Luo,

We are pleased to inform you that your manuscript 'Large language model detects previously undiagnosed heart failure with preserved ejection fraction in patients with metabolic-associated fatty liver disease: a multicenter cohort study' has been provisionally accepted for publication in PLOS Digital Health.

Best regards,

Thomas Kyumwa Kisimbi, RN, MA, MPA, MBA

Academic Editor

PLOS Digital Health

**Additional Editor Comments (if provided):**

All comments have been addressed. There is a minor mistake that needs to be corrected. The author have provided the section "Study design and data sources". The first paragraph of "Ethical considerations" needs to be removed.

**Reviewer Comments (if any, and for reference):**

Reviewer's Responses to Questions

**Comments to the Author**

Reviewer #1: All comments have been addressed

Reviewer #2: All comments have been addressed

Reviewer #3: All comments have been addressed

publication criteria ? Is the manuscript technically sound, and do the data support the conclusions? The manuscript must describe methodologically and ethically rigorous research with conclusions that are appropriately drawn based on the data presented.

Reviewer #1: Yes

Reviewer #2: Yes

Reviewer #3: Yes

3. Has the statistical analysis been performed appropriately and rigorously?

Reviewer #1: Yes

Reviewer #2: Yes

Reviewer #3: Yes

4. Have the authors made all data underlying the findings in their manuscript fully available (please refer to the Data Availability Statement at the start of the manuscript PDF file)?

Reviewer #1: Yes

Reviewer #2: Yes

Reviewer #3: Yes

5. Is the manuscript presented in an intelligible fashion and written in standard English?

Reviewer #1: Yes

Reviewer #2: Yes

Reviewer #3: Yes

Reviewer #1: Thank you for the thorough revision and the detailed, point-by-point responses to the reviewers’ prior comments. From a language and presentation perspective, this revision is clearly more polished and easier to read.

In particular, I appreciate the authors’ concrete editing work to improve consistency and reduce ambiguity across the manuscript, including: (i) standardizing model naming/formatting throughout (e.g., consistent use of “ChatGPT-4,” “Llama 3.1 (405B),” and “MedGuide-14B” and consistent punctuation around model variants); (ii) harmonizing in-text citation formatting (e.g., replacing superscript-style citations with bracketed numerical references); and (iii) cleaning up manuscript-wide typography issues (e.g., replacing full-width/non-ASCII punctuation with standard ASCII punctuation, normalizing percentage formatting, and standardizing units/notation). These changes materially improve readability and help ensure the text aligns with journal style requirements.

I also found several wording edits helpful for clarity and interpretation (e.g., clarifying the description of imaging-related data use in the Abstract, and revising potentially misleading phrasing/headings such as the “Prospective Validation” wording to more accurately reflect the validation design). Overall, the manuscript is presented in an intelligible fashion and is written in standard English suitable for publication.

Only minor remaining copy-editing is recommended at this stage (a final proofreading pass to catch any lingering small typos/spacing/encoding artifacts, especially in figure captions/legends), but no major language revision appears necessary.

Reviewer #2: All comments have been addressed. There is a minor mistake that needs to be corrected. The author have provided the section "Study design and data sources". The first paragraph of "Ethical considerations" needs to be removed.

Reviewer #3: The authors have provided a thorough and high-quality response to all the comments raised in the initial review. In summary, all my concerns have been addressed. The manuscript is now well-prepared for publication in PLOS Digital Health.

**Do you want your identity to be public for this peer review?** For information about this choice, including consent withdrawal, please see our Privacy Policy .

Reviewer #1: No

Reviewer #2: No

Reviewer #3: No
